# Manufacturing of high strength and high conductivity copper with laser powder bed fusion

Yingang Liu [1,11], Jingqi Zhang [1,11], Ranming Niu [2,3] ✉, Mohamad Bayat[4], Ying Zhou[5], Yu Yin [1], Qiyang Tan [1], Shiyang Liu[1], Jesper Henri Hattel[4], Miaoquan Li[6], Xiaoxu Huang [7,8], Julie Cairney [2,3], Yi-Sheng Chen [2,3], Mark Easton [9], Christopher Hutchinson [10] ✉ & Ming-Xing Zhang [1] ✉

Additive manufacturing (AM), known as 3D printing, enables rapid fabrication of geometrically complex copper (Cu) components for electrical conduction and heat management applications. However, pure Cu or Cu alloys produced by 3D printing often suffer from either low strength or low conductivity at room and elevated temperatures. Here, we demonstrate a design strategy for 3D printing of high strength, high conductivity Cu by uniformly dispersing a minor portion of lanthanum hexaboride ($LaB_6$) nanoparticles in pure Cu through laser powder bed fusion (L-PBF). We show that trace additions of $LaB_6$ to pure Cu results in an improved L-PBF processability, an enhanced strength, an improved thermal stability, all whilst maintaining a high conductivity. The presented strategy could expand the applicability of 3D printed Cu components to more demanding conditions where high strength, high conductivity and thermal stability are required.

The ability to fabricate fully dense, highly conductive, and mechanically robust copper (Cu) components is essential for electrical conduction and heat management applications. Additive manufacturing (AM)[1–11], or 3D printing, offers unprecedented opportunities for producing Cu components with complex geometry and tailored performance which are inaccessible by conventional manufacturing processes. However, pure Cu has a high reflectivity for infrared lasers and hence pure Cu components fabricated by the most commonly used laser AM machines often suffers from high porosity[12–14], leading to poor mechanical and conductivity properties. Although AM using green lasers or electron beams enables the fabrication of pure Cu

components with relatively high density[15–17], the intrinsically low strength of pure Cu at room temperature and its inability to resist thermal softening hinders the applications of additively manufactured Cu components in demanding mechanical loading and high temperature conditions.

Alloying Cu with elements such as Cr and Zr improves the laser absorptivity and strengthens the metal[12,18,19], but this approach degrades the conductivity due to their high solid solubility in Cu[20]. An alternative approach is to add immiscible foreign particles to reinforce the Cu while maintaining the high conductivity[21,22]. In practice, it has proven exceedingly difficult to achieve a sufficiently well dispersed

[1]School of Mechanical and Mining Engineering, The University of Queensland, St. Lucia, QLD, Australia. [2]Australian Centre for Microscopy and Microanalysis, The University of Sydney, Sydney, NSW, Australia. [3]School of Aerospace, Mechanical and Mechatronic Engineering, The University of Sydney, Sydney, NSW, Australia. [4]Department of Mechanical Engineering, Technical University of Denmark, Lyngby, Denmark. [5]State IJR Center of Aerospace Design and Additive Manufacturing, Northwestern Polytechnical University, Xi'an, China. [6]School of Materials Science and Engineering, Northwestern Polytechnical University, Xi'an, China. [7]International Joint Laboratory for Light Alloys (Ministry of Education), College of Materials Science and Engineering, Chongqing University, Chongqing, China. [8]Shenyang National Laboratory for Materials Science, Chongqing University, Chongqing, China. [9]Centre for Additive Manufacturing, School of Engineering, RMIT University, Melbourne, VIC, Australia. [10]Department of Materials Science and Engineering, Monash University, Clayton, VIC, Australia. [11]These authors contributed equally: Yingang Liu, Jingqi Zhang. ✉e-mail: ranming.niu@sydney.edu.au; christopher.hutchinson@monash.edu; mingxing.zhang@uq.edu.au

and uniform distribution of externally added nanoparticles to obtain a significant strengthening increment, without particle agglomeration degrading the ductility and damage tolerance[21,23]. The challenge of simultaneously obtaining high strength and high conductivity in 3D printed Cu parts limits its applications where a good balance of mechanical and physical properties is required.

Here, we present a design strategy that enables laser AM of Cu parts with high density and high performance through the addition of minute amounts of lanthanum hexaboride ($LaB_6$) nanoparticles. Our $LaB_6$-doped Cu exhibits a yield strength of $347 \pm 2$ MPa (which is 3.7 times higher than pure Cu), together with an elongation to failure of $22.8 \pm 1.2\%$, electrical conductivity of 98.4% IACS (International Annealed Copper Standard), thermal conductivity of 387 W m$^{-1}$ K$^{-1}$ and the ability to resist softening at 1050 °C. Additionally, we further demonstrate the applicability of our design strategy to geometrically complex components.

## Results

### Design strategy

The key feature of our design strategy is that we select additives, of which the constituent elements have negligible solubility in solid Cu (and therefore have negligible detrimental effects on the conductivity), but will dissolve in the melt pools during laser melting (therefore not too high melting point) and subsequently re-precipitate with a very fine dispersion during solidification (and hence provide excellent strengthening). Criteria are as follows:

(1)  the constituent elements of the particles should have negligible solid solubility in Cu in order to minimize their detrimental influence on the conductivity[19,20,24,25] and to maximize the re-precipitation of nanoparticles;

(2)  particles should have a relatively low melting point[26], to maximize the chance of dissolution in the melt pool and to minimise coarsening of the re-precipitated nanoparticles during solidification; and

(3)  particles should possess a low wetting angle with liquid Cu[27], minimizing nanoparticle clustering in the liquid Cu.

## Microstructural characterization

Following this design strategy, we identified $LaB_6$ as a suitable additive candidate. $LaB_6$ met the criteria due to its lower melting point, minimal solid solubility of constituent elements in Cu, and low wetting angle with molten Cu (see Supplementary Note 1). The initial $LaB_6$ particles that were added to the pure Cu powder feedstock exhibited an irregular morphology and sizes up to 300 nm (Supplementary Fig. 1). The laser reflectivity test clearly demonstrates that pure Cu powder exhibits exceptionally high laser reflectivity within the infrared laser range (Fig. 1b), specifically at a wavelength of 1060 nm, as used by the laser powder bed fusion (L-PBF) system in our study. The reflectivity was noticeably decreased upon the incorporation of 1.0 weight per cent (wt%) $LaB_6$ nanoparticles. This reduction can be attributed to two factors: the inherently low laser reflectivity of $LaB_6$ and the introduction of $LaB_6$ nanoparticles, which enhance the surface roughness of pure Cu particles, facilitating multiple reflections within the powder bed.

We then produced pure Cu parts using L-PBF and compared them with parts produced from Cu with 1 wt% $LaB_6$ nanoparticles (hereafter denoted as 1.0$LaB_6$-Cu) (see Methods). Despite the relatively high energy density being used for laser melting, the pure Cu parts produced by L-PBF shows discontinuous scanning tracks with a rough surface (Fig. 1a$_2$), indicating a balling effect resulting from the high reflectivity of infrared lasers[12,13,28]. This leads to lack-of-fusion defects when successive layers are fused[29]. Micro-computed tomography (Micro-CT) shows pores with a diameter range of 10-130 μm (Supplementary Fig. 2a, b). In contrast, the 1.0$LaB_6$-Cu part demonstrates well-defined scanning tracks and no obvious defects (Fig. 1a$_3$), which were attributed to the increased laser absorptivity due to the $LaB_6$ addition (Fig. 1b). Micro-CT and electron back-scatter diffraction (EBSD) confirm the much higher density (Supplementary Fig. 2e, f) and relatively large grains of the 1.0$LaB_6$-Cu as compared with pure Cu (Fig. 1a$_3$). With an improvement in laser processability, highly dense 1.0$LaB_6$-Cu parts can be fabricated effectively over a broader laser power range, spanning 325 W to 400 W (Supplementary Fig. 3).

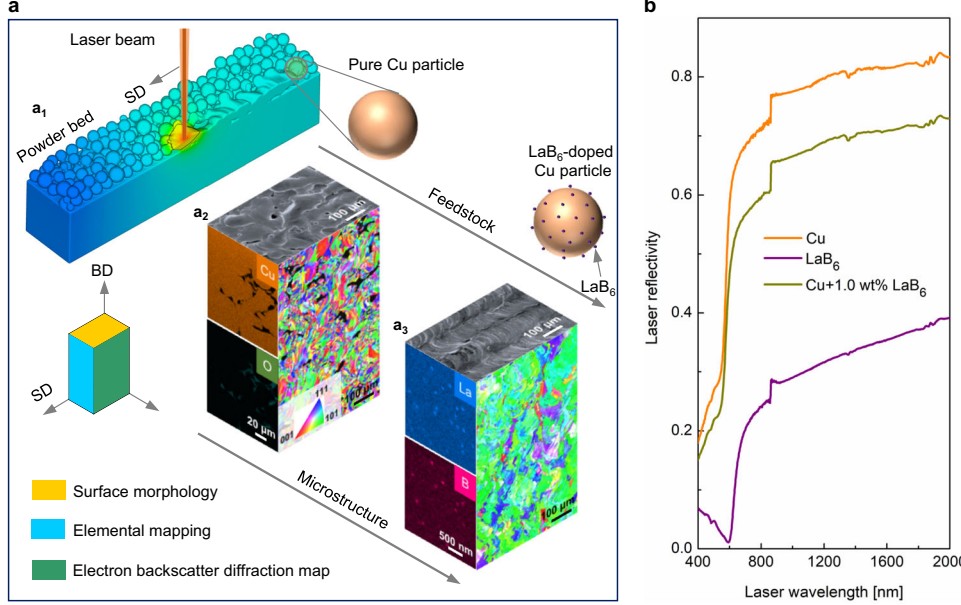

**Fig. 1 | Laser AM of Cu and $LaB_6$-doped Cu. a** A schematic diagram of laser AM process (**a$_1$**), and top surface morphology, cross-sectional EDS elemental mapping and EBSD inverse pole figure map of the L-PBF fabricated pure Cu (**a$_2$**) and 1.0$LaB_6$-Cu (**a$_3$**). The pores are visible in pure Cu and high density can be achieved after addition of $LaB_6$. **b** Laser reflectivity of pure Cu, $LaB_6$ and 1.0 wt% $LaB_6$ nanoparticles doped Cu powder feedstock at various laser wavelengths. After introducing 1.0 wt% $LaB_6$ nanoparticles, the powder mixture shows lower laser reflectivity compared with pure Cu powder, indicating the increased laser absorptivity. SD Scanning direction, BD Building direction. Source data are provided as a Source Data file.

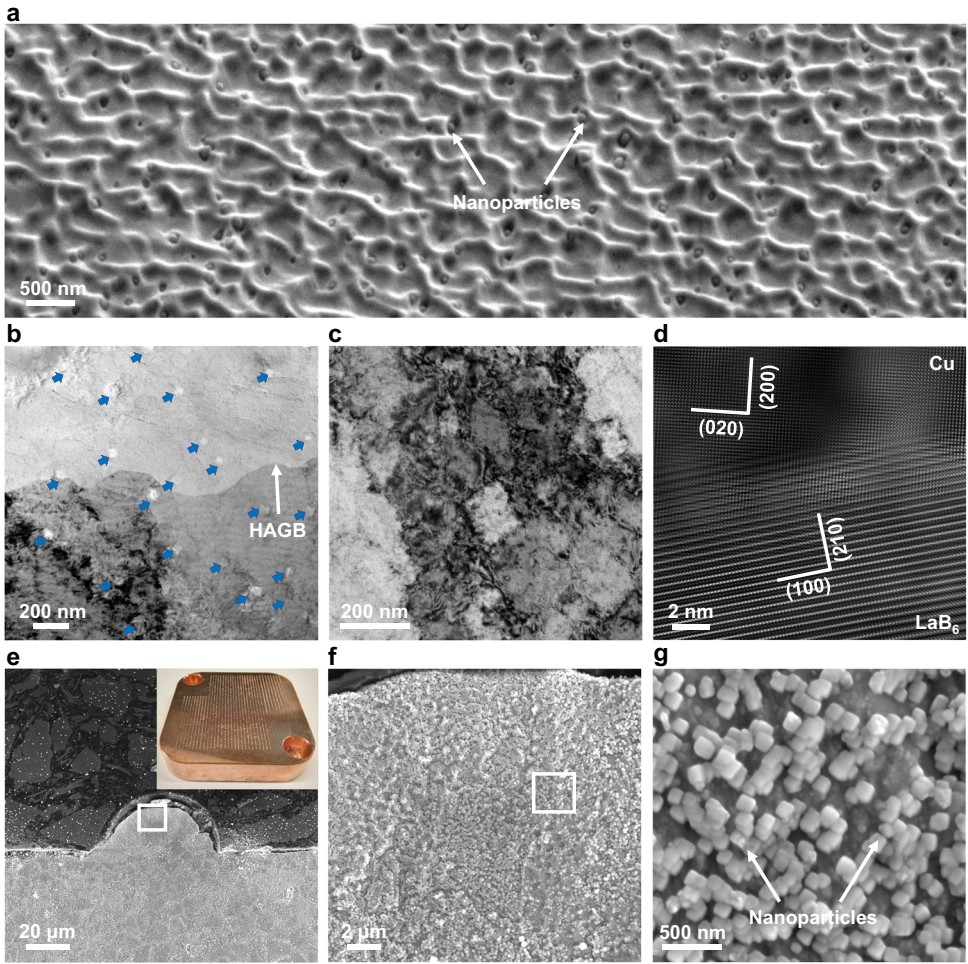

**Fig. 2 | Characterization of LaB$_6$ nanoparticles. a** SEM image showing the homogeneous distribution of LaB$_6$ nanoparticles within one Cu grain in the L-PBF fabricated 1.0LaB$_6$-Cu. **b** Bright-field TEM image of LaB$_6$ nanoparticles across a high angle grain boundary (HAGB). No particle agglomeration is found. **c** High magnification bright-field TEM image showing the morphology of the LaB$_6$ nanoparticles. Note that the initial irregular shape of the nanoparticles in the powder feedstock disappeared and rectangular-shaped LaB$_6$ nanoparticles are observed. **d** Fourier-filtered high-resolution TEM image showing the interface between the LaB$_6$ nanoparticle and Cu matrix. **e** SEM image of single-track sample. **f, g** Higher magnification SEM images taken from the frame regions in (**e, f**) respectively. Large number of rectangular-shaped nanoparticles are visible in the single-track sample which is free from thermal cycling during L-PBF, demonstrating that these nanoparticles formed in the Cu melt during solidification.

Scanning electron microscopy (SEM) - energy-dispersive X-ray spectroscopy (EDS) mapping and X-ray diffraction (XRD) reveal that only LaB$_6$ nanoparticles are identified in the as-fabricated 1.0LaB$_6$-Cu part (Fig. 1a$_3$, Supplementary Fig. 4a, b). We examined the LaB$_6$ nanoparticles using a focused ion beam (FIB) SEM at a 52° tilt, which allows for exposure of nanoparticles on the Cu matrix surface. The LaB$_6$ nanoparticles are uniformly distributed within the grains without agglomeration (Fig. 2a). We further characterized the nanoparticles in transmission electron microscopy (TEM), which confirms the uniform dispersion of LaB$_6$ (Fig. 2b, see details in Supplementary Note 2). The LaB$_6$ nanoparticles exhibit a rectangular shape with an average size below 100 nm, and an incoherent interface with the Cu matrix (Fig. 2b–d).

### Elemental analysis

We subsequently performed compositional analysis of the 1.0LaB$_6$-Cu using atom probe tomography (APT). Only La and B clusters were identified. Although the atomic ratio of B to La of the clusters appears less than 6:1 (Fig. 3) due to the formation of surface boron hydride[30] and the trajectory aberration[31] caused by the different evaporation fields of the nanoparticles and Cu matrix, the APT data approximately shows the formation of nanoparticles rather than dissolved La and B

solute in the Cu matrix (Fig. 3, Supplementary Fig. 5, See Supplementary Note 3). It indicates that any dissolution of La and B in the Cu matrix is very limited, and this minimizes any detrimental effect of solid solution solutes on the conductivity. This is further supported by the XRD analysis, which shows that the lattice parameter of the Cu matrix is the same as pure Cu, suggesting that negligible La or B are dissolved in the solid Cu matrix (Supplementary Fig. 4c). In general, the APT elemental analysis supports our hypothesis that La and B tend to in situ form La–B nanoparticles as negligible La and B were detected in the Cu matrix except for the La and B enrichment regions. The XRD (Supplementary Fig. 4) and TEM (Fig. 2d) results confirm the nanoparticles are LaB$_6$. It can be concluded that the externally added LaB$_6$ particles that were irregular in shape and larger in size have dissolved in the melt pools during heating and the observed LaB$_6$ nanoparticles are the product of re-precipitation during solidification, as designed.

### Evidence of re-precipitation during solidification

To prove that the re-precipitation of LaB$_6$ nanoparticles directly occurs during solidification and to exclude the possibility of solid-state phase transformation induced by thermal cycling, we performed additional single-track experiment using the 1.0LaB$_6$-Cu feedstock and the same

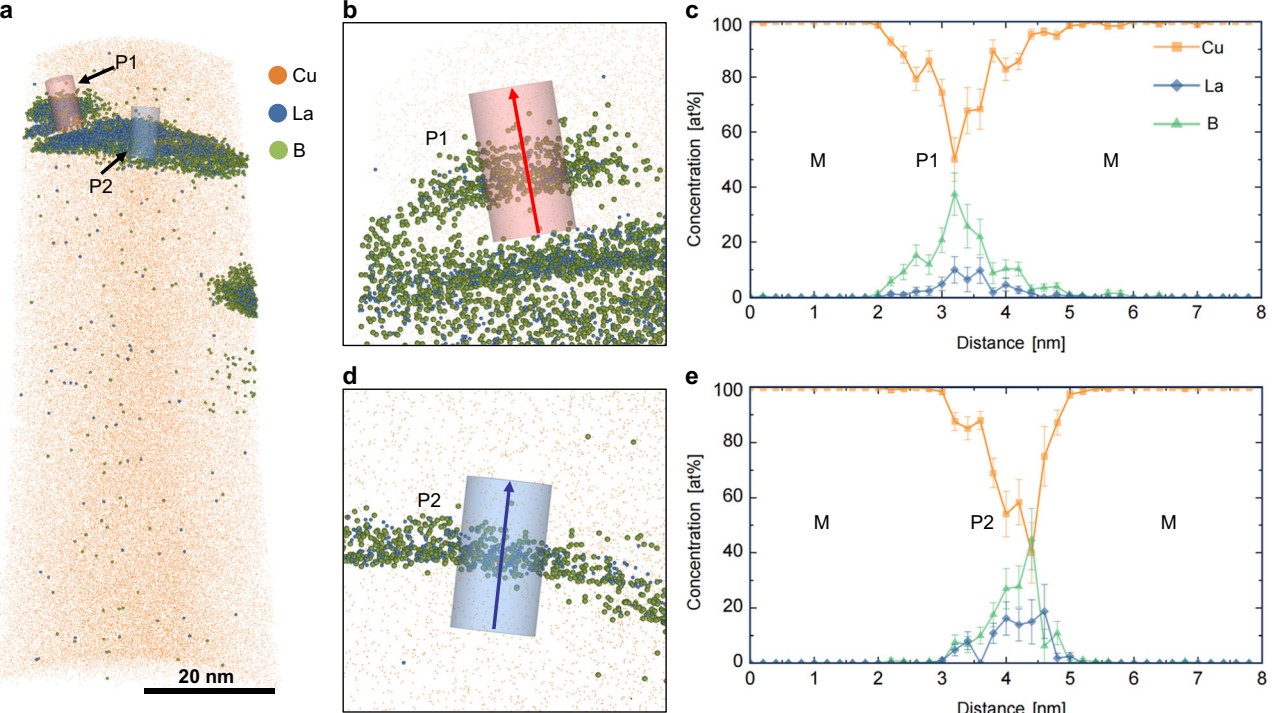

**Fig. 3 | Atom probe tomography (APT) characterization of 1.0LaB₆-Cu. a** 3D reconstruction of Cu, La and B distribution in the L-PBF fabricated 1.0LaB₆-Cu. **b** Enlarged view of the precipitate−1 (P1). **c** The 1-D concentration profile along the axis of the cylindrical volume-of-interest in (**b**). **d** is the enlarged view of P2 and (**e**) is the 1-D concentration profile crossing P2. Both concentration profiles showing depletion of La and B in Cu matrix. La enrichment is spatially correlated with B. Note that the La and B enriched zones do not correspond to the real shape and dimension of the LaB₆ nanoparticles due to the significant different voltages required for the evaporation of LaB₆ and Cu during APT (see another dataset in Supplementary Fig. 5). The error bars denote the standard deviation of the mean. M: Matrix, P: Precipitate. Source data are provided as a Source Data file.

L-PBF processing parameters as the bulk sample. Despite the fact that the size and density of LaB₆ nanoparticles in the single-track sample differ from those in the bulk sample due to layer-layer and track-track laser remelting that can potentially modify the size and density in the bulk sample, large number of rectangular-shaped nanoparticles are visible in the single-track sample (Fig. 2e–g). This confirms the re-precipitation of LaB₆ nanoparticles during solidification in the L-PBF process because there was no thermal cycling effect on the single-track sample to facilitate the precipitation from the solid Cu. In addition, our crystallographic examination does not reveal any reproducible orientation relationship between the LaB₆ nanoparticles and the Cu matrix (Fig. 2d, Supplementary Fig. 6), which indicates an incoherent interface. If the LaB₆ nanoparticles precipitated from the solid Cu, they would display either coherent or semi-coherent interfaces with the matrix to lower the interfacial energy. As such, reproducible orientation relationships would be generally observed[32,33]. Therefore, the LaB₆ nanoparticles precipitated directly from the Cu melt during solidification rather than from solid-state phase transformations. To further verify this key conclusion, we repeated the experiment using micron-size LaB₆ particles as the additive. All the LaB₆ microparticles disappeared and nanoparticles are observed (Supplementary Fig. 7a, b), demonstrating that the observed LaB₆ nanoparticles are not the initial additives, but the product of re-precipitation from the liquid Cu during solidification after dissolving during laser melting. In addition, we also examined the top surface of the L-PBF fabricated 1.0LaB₆-Cu sample, where there is no extensive thermal cycling, and as we observed in other areas, we also found a large number of uniformly distributed LaB₆ nanoparticles (Supplementary Fig. 8). This observation, coupled with the results from the single-track experiment, suggests that the proposed design strategy is suitable for the geometrically complex components with fine structures due to the formation of uniformly dispersed nanoparticles during solidification.

## Mechanical and conductivity properties

To evaluate the effect of the LaB₆ nanoparticles on the mechanical and conductivity properties, we performed tensile tests and electrical conductivity measurements on both the L-PBF fabricated pure Cu and 1.0LaB₆-Cu. Pure Cu delivers a yield strength of $73 \pm 2$ MPa, ultimate tensile strength of $121 \pm 1$ MPa and an elongation to failure of $10.8 \pm 1.1\%$ (Fig. 4a), with an electrical conductivity of 88.3% IACS (Fig. 4b). The corresponding thermal conductivity is calculated to be 347 W m⁻¹ K⁻¹ based on the Wiedemann-Franz law (Methods). The low strength and conductivity of the pure Cu are due to the presence of the high density of pores (Fig. 1a₂, Supplementary Fig. 2a, b). The 1.0LaB₆-Cu exhibits a yield strength of $347 \pm 2$ MPa, ultimate tensile strength of $412 \pm 7$ MPa and elongation to failure of $22.8 \pm 1.2\%$ – with an electrical conductivity of 98.4% IACS (Fig. 4a, b). The thermal conductivity is calculated to be 387 W m⁻¹ K⁻¹. While it is possible to create highly dense pure Cu parts through AM using green laser, the strength of these parts remains significantly lower than that achieved in our study[15,16]. The significant strength enhancement is mainly attributed to dispersion strengthening from the LaB₆ nanoparticles (Supplementary Note 4). The ductility improvement is partially due to the improvement in strain hardening resulting from the uniform dispersion of shear-resistant nanoparticles, and to the reduced porosity and defects. The conductivity improvement arises from the higher density of the as-fabricated part and the minimal negative effects of the uniformly dispersed LaB₆ nanoparticles. Overall, high strength and high ductility combined with high conductivity make the 1.0LaB₆-Cu superior to the conventionally and additively manufactured Cu, Cu alloys and Cu matrix composites[12,13,19,21,22,24,25,34–43] (Fig. 4b, Supplementary Note 5). For example, the 1.0LaB₆-Cu shows a good balance of strength and ductility, and much higher electrical conductivity compared with Cu alloys produced by L-PBF, which typically require a post-AM annealing to restore the electrical conductivity[12,19,24,25,39,40]. Even if Glenn Research

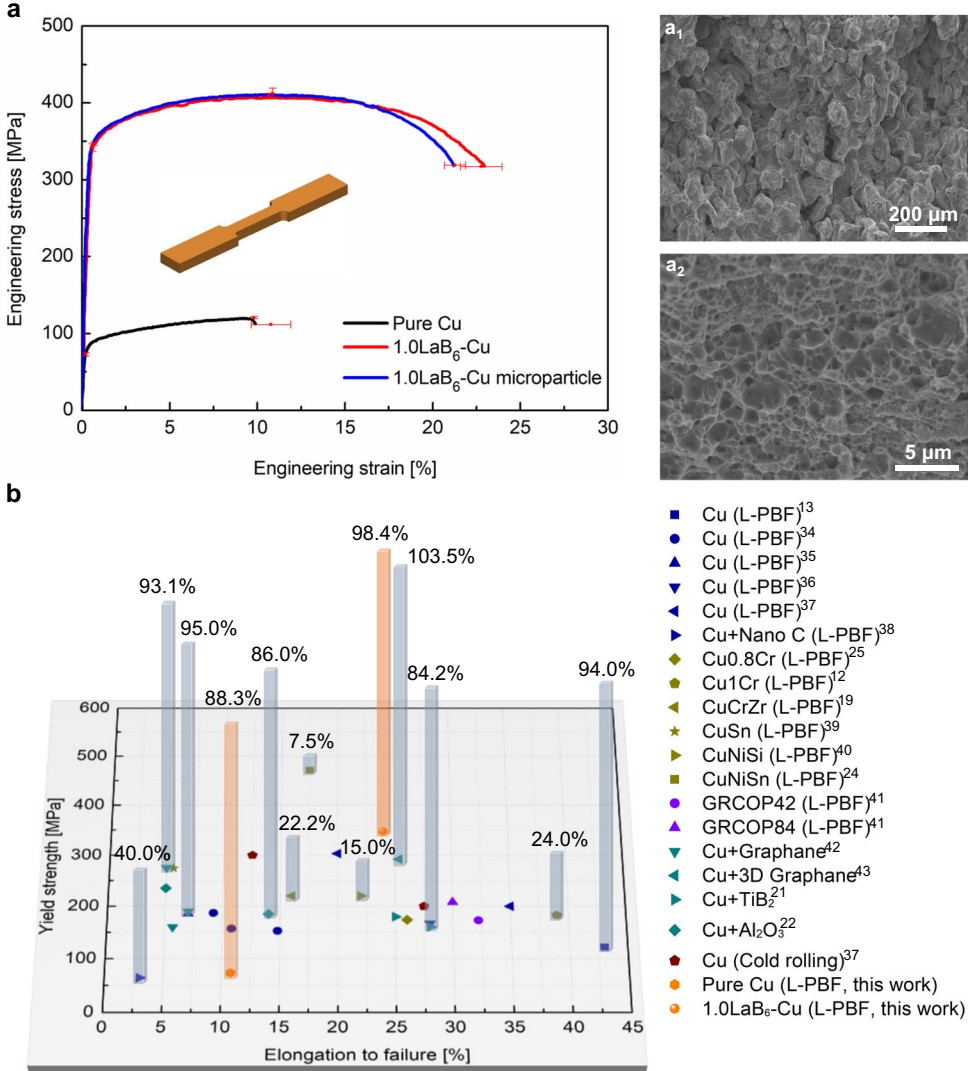

**Fig. 4 | Tensile mechanical property and electrical conductivity. a** Tensile engineering stress-strain curves and the fracture surface morphology of the L-PBF fabricated pure Cu (**a₁**) and 1.0LaB₆-Cu (**a₂**). The error bars represent the standard deviation of the mean. **b** Comparison of the yield strength, elongation to failure and electrical conductivity obtained in the present work with previously published data of high-performance Cu, Cu alloys and Cu matrix composites fabricated by L-PBF, L-PBF plus heat treatment and traditional processing[12,13,19,21,22,24,25,34–43]. Note that the columns and corresponding values represent the electrical conductivity (IACS, International Annealed Copper Standard). Source data are provided as a Source Data file.

Copper 42 (GRCop-42) alloy and GRCop-84 alloy, the niobium chromide (Cr₂Nb) precipitation hardened alloys, have shown suitability for AM, their conductivity of 85% IACS and 75% IACS is still lower than the 1.0LaB₆-Cu due to the solid solution of Cr and Nb in the Cu matrix[41]. In addition to the higher conductivity, the 1.0LaB₆-Cu also exhibits higher strength compared to GRCop-42 and GRCop-84 alloys, with the latter alloys demonstrating a yield strength within the range of 172 to 208 MPa[41]. Furthermore, although 1.0LaB₆-Cu shows comparable electrical conductivity to Cu matrix composites produced by conventional processes[21,22], its unique combination of mechanical properties, coupled with the greater design freedom afforded by AM, is attractive for practical applications where mechanically robust, electrically/thermally conductive as well as geometrically complex Cu components are in demand. It has been observed that graphene-reinforced Cu matrix composites show slightly higher electrical conductivity and ductility[43]. However, the L-PBF produced 1.0LaB₆-Cu can achieve higher strength and geometric complexity, at lower cost owing to the small addition level and much lower cost of LaB₆. In addition, the L-PBF fabricated 1.0LaB₆-Cu using LaB₆ microparticles as additive to the feedstock also shows comparable mechanical response (Fig. 4a).

In addition to the room-temperature mechanical properties, the 1.0LaB₆-Cu also shows improved thermal stability at elevated temperatures. Cu and its alloys are often used in elevated temperature environments (heating due to electrical currents, or heat management applications)[44]. After exposure to elevated temperatures, they may suffer from substantial strength loss, which can eventually lead to service failures. Unlike Cu alloys and other oxide dispersion-strengthened Cu which show relatively low softening temperatures (typically below 600 °C) due to coarsening of the strengthening particles[45,46], the 1.0LaB₆-Cu part shows exceptional resistance to softening up to 1050 °C (Supplementary Fig. 9). Even after annealing at 1050 °C, 80% of the ultimate tensile strength is retained together with an elongation to failure of 28% (Supplementary Note 6).

As an example of a geometrically complex Cu component fabricated by 3D printing, we show the ability of the LaB₆-doped Cu to bear compression loading in sheet-based gyroid lattice structures fabricated with L-PBF using both pure Cu and LaB₆-doped Cu powder feedstock (Methods). The pure Cu gyroid structure shows an inferior mechanical response to loading and undergoes dramatic post-yield softening at a strain range of 15–20% (Fig. 5a), due to the formation of

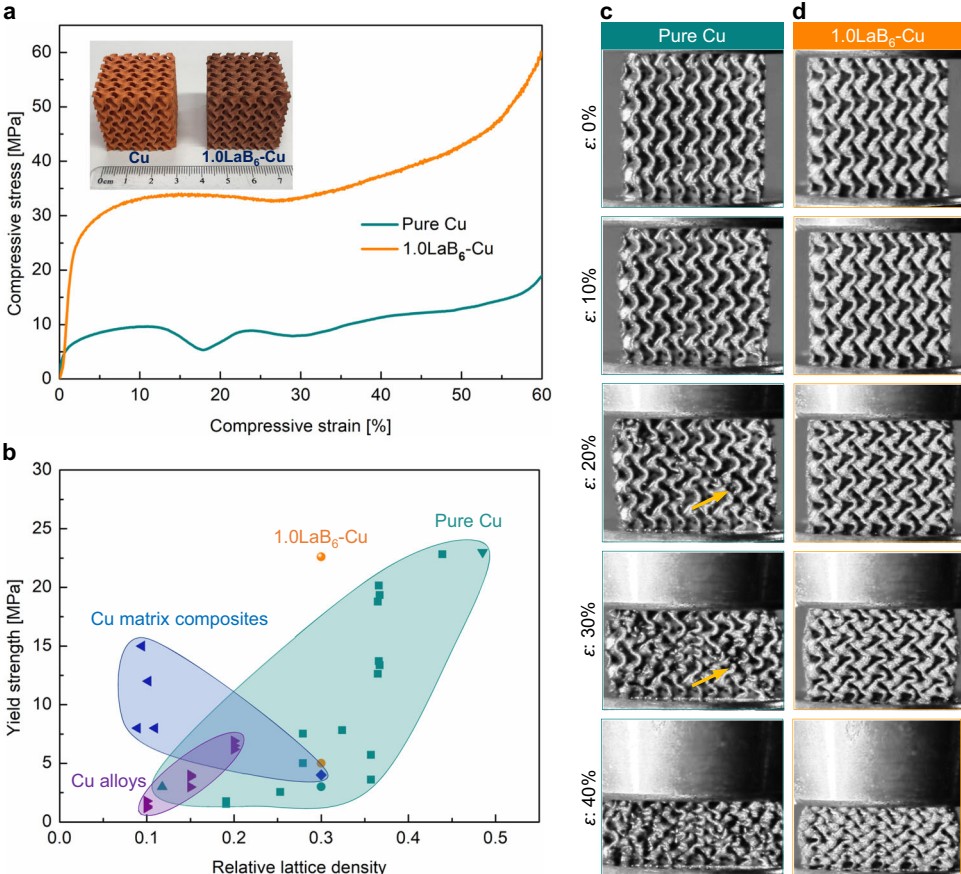

**Fig. 5 | Compressive mechanical property of geometrically complex components. a** Compressive engineering stress-strain curves of the L-PBF fabricated sheet-based gyroid lattices made of pure Cu and 1.0 wt% LaB$_6$ nanoparticles doped Cu powder feedstocks. **b** Comparison of the compressive yield strength against relative lattice density between the present results and previously published data of lattices made of Cu, Cu alloys and Cu based composites[47–51]. **c** Video camera frames during compression of pure Cu lattice. **d** Video camera frames during compression of 1.0LaB$_6$-Cu lattice. The pure Cu lattice showed development of localized shear bands (identified by the arrows), which were not evident in 1.0LaB$_6$-Cu lattice. Source data are provided as a Source Data file.

localized shear bands (Fig. 5c). In contrast, the addition of LaB$_6$ to Cu not only enhances the yield strength, but also resists shear band localization (Fig. 5d), albeit a slight softening behavior occurs at high strains (between 25% and 30%) (Fig. 5a). Figure 5b summarizes the compressive yield strength of porous Cu parts versus relative lattice density[47–51]. The lattice made of LaB$_6$-doped Cu has a yield strength of 23 MPa at a relative lattice density of 0.3, which is 3.5 times higher than the strength of the same lattice made of pure Cu of the same relative lattice density. This yield strength is comparable to other porous pure Cu lattices that have a much higher relative lattice density (0.439 and 0.485)[47]. This property is attractive for applications where the total mass or volume of lattice structures is the critical design criterion. Overall, by combining the effect of LaB$_6$ in Cu with the design freedom afforded by AM, we achieve high strength and high conductivity, and can create geometrically complex components with the increased ability to resist compressive loading.

In summary, we have demonstrated a pathway for the reliable fabrication of highly dense Cu parts with high strength and high conductivity through minor addition of LaB$_6$ particles into the pure Cu feedstock for 3D printing. The key to the approach is to introduce appropriate particles to pure Cu, which can dissolve in the melt pool and re-precipitate uniformly during solidification (Supplementary Note 7). This newly developed 1.0LaB$_6$-Cu by L-PBF fills a major gap in 3D printing of metal alloys and can be used in applications where demanding mechanical and electrically/thermally conductive properties are required. Since uniformly dispersed nanoparticles are often used to strengthen metallic materials, this design strategy and the

associated dissolution and reprecipitation during solidification could be extended to other alloy systems for the development of use-as-printed high-strength materials for AM.

## Methods

### Laser powder bed fusion

Gas atomized oxygen-free Cu powder with a purity of at least 99.95 wt %, globally spherical in shape and particle size ranging from 15 μm to 53 μm was used as the feedstock. The Cu powder exhibits D10, D50, and D90 values of 15 μm, 36 μm, and 54 μm, respectively. High purity (99.9%) LaB$_6$ nanoparticles with an average size of 100 nm (the maximum size was up to 300 nm) was used as the additive (Supplementary Fig. 1a, b). The LaB$_6$ powder demonstrates D10, D50, and D90 measurements of 54 nm, 85 nm, and 145 nm, respectively. In this work, LaB$_6$ nanoparticles were first ultrasonically vibrated for 0.5 h using Retsch® UR 1 to dissociate agglomerates. Subsequently, LaB$_6$ nanoparticles were mechanically mixed with pure Cu powder in a Turbula mixer for 0.5 h to ensure a homogeneous distribution[52,53]. The addition level was optimized based on part density and tensile testing results. After building samples with 0.5–2.0 wt% additions, we found that 1.0 wt% addition was the lowest addition to ensure the density over 99.5% measured using micro-computed tomography (micro-CT) analysis (Supplementary Fig. 2). This was further confirmed through Archimedes' method. In addition, this addition level also corresponds to the optimal strength and ductility compared to the lower addition of 0.5 wt% and higher addition of 2.0 wt% (Supplementary Fig. 7c). In contrast to 1.0LaB$_6$-Cu, high density cannot be achieved in 0.5LaB$_6$-Cu

and the nanoparticle agglomeration was observed in the 2.0LaB$_6$-Cu due to the excessive addition (Supplementary Fig. 7d), leading to a reduction in the strength and ductility. After the feedstock preparation, the powder mixture was characterized in a SEM (Hitachi SU3500) equipped with EDS. LaB$_6$ nanoparticles electrostatically adhere to almost all Cu particles with uniform distribution (Supplementary Fig. 1). As a comparison to the experiment, high purity (99.9%) TiB$_2$ nanoparticles with an average size of 100 nm were also mixed with pure Cu powder (Supplementary Fig. 13a). Furthermore, micrometre-scaled LaB$_6$ powder with particle size ranging from 1 μm to 10 μm was also employed as the additive and mixed with pure Cu powder to show the microstructure and property response (Supplementary Fig. 7a). The laser reflectivity of the as-received powder and powder mixtures was measured using a double-beam UV-Visible-NIR Lambda 1050 PerkinElmer spectrophotometer equipped with a 150 mm integrating sphere in the 400–2000 nm wavelength range. Pure Cu powder and the powder mixtures were subsequently subjected to laser powder bed fusion (L-PBF) AM in an SLM125HL system and the processing parameters are listed in Supplementary Table 1. The L-PBF fabricated dogbone-shaped blocks with gauge length of 10 mm, gauge width of 2.5 mm, and thickness of 40 mm were used for tensile testing[54]. The chemical compositions of the L-PBF fabricated samples were measured using inductively coupled plasma atomic emission spectroscopy (ICP-AES) for metallic elements and using LECO combustion analysis for non-metallic elements, as listed in Supplementary Table 2. Moreover, in order to analyse the microstructure of directly solidified metal, single-track samples were prepared with the same L-PBF processing parameters as the bulk sample on a pure Cu substrate (Fig. 2e). In addition, the sheet-based gyroid lattice with porosity of 70% and overall dimension of 30 mm (length) × 30 mm (width) × 30 mm (height) was designed to contain 5 × 5 × 5 arrays of unit cells and fabricated by L-PBF using both pure Cu and LaB$_6$ nanoparticles doped Cu powder feedstocks for compression tests (Fig. 5a).

## Heat treatment

In order to show the thermal stability of the L-PBF fabricated 1.0LaB$_6$-Cu, annealing treatments were performed at various temperatures ranging from 550 °C to 1050 °C for 1 h in a vacuum furnace, followed by furnace cooling at a cooling rate of 5 °C min$^{-1}$ to ambient temperature. Hardness of heat-treated samples was measured in a LECO Vickers hardness tester at a load of 3 kg and a dwell time of 12 s. Six random positions were tested to obtain the average value for each sample.

## Microstructure characterization

All metallographic samples were firstly ground using SiC papers followed by polishing with Struers OP-S suspension and finally etching with 50 ml HCl, 20 ml Fe$_3$Cl, and 30 ml C$_2$H$_5$OH for 3 s. The microstructure was examined in a Hitachi SU3500A SEM. Grain size, grain morphology and crystallographic orientation were characterized through electron backscatter diffraction (EBSD) analysis at a scan step size of 1.2 μm in a JEOL JSM-7800F field emission SEM and the corresponding geometrically necessary dislocation (GND) density was calculated. The detailed description of the method used for GND density calculation can be found in ref. [55]. Prior to EBSD, surfaces of samples were electropolished in a Struers electrolyte D2. In addition, typical element distributions in the samples were analysed using EDS in the SEM operated at an accelerating voltage of 20 kV. To examine the nanoparticle distribution, the electropolished samples were etched with gallium ions using a focused ion beam (FIB), and SEM images were acquired at a 52° tilt to expose the nanoparticles in a FEI Scios FIB-SEM. Phase analysis of the L-PBF fabricated samples was accomplished using XRD in a Brücker D8 Advance system at a scan rate of 1° min$^{-1}$ and step size of 0.02° with the diffraction angle 2θ ranging from 20° to 100°. FIB was used to prepare the TEM thin foils. A Hitachi HF5000 TEM, equipped with a probe aberration corrector and symmetrically opposed dual EDS detectors and operated at an acceleration voltage of 200 kV, was used for in-depth characterization of the L-PBF fabricated 1.0LaB$_6$-Cu. The diondo d2 micro-CT system was used to analyse the micropores in the L-PBF fabricated samples. 2D slice image data were acquired through an X-ray source with the voltage of 120 kV and current of 100 μA. The exposure time was 2000 ms with a spatial solution of 4 μm.

## Atom probe tomography

The nanoscale elemental distribution in the L-PBF fabricated 1.0LaB$_6$-Cu was analysed using APT in a local electrode atom probe CAMECA LEAP 4000X SI, which has a detection rate of 57%. APT matchstick specimens with dimension of 1.0 mm × 1.0 mm × 10 mm were first cut from the as-built sample, and then electropolished using alternating current under voltages of 5 V, 3 V and 1 V. The electrolyte was 70% orthophosphoric acid in water, as suggested by ref. [31]. The electropolished APT samples were then annular polished using a Thermo Fisher G4 Plasma FIB-SEM. A Xe$^+$ beam with a voltage of 5 kV and a current of 30 pA were used to finalize the APT sample preparation. The laser APT experiment was conducted under a high vacuum of $2 \times 10^{-11}$ torr, at a temperature of 50 K, a laser pulse frequency of 200 kHz, and a laser pulse energy of 70 pJ. The APT data was reconstructed using a CAMECA integrated visualization and analysis software (AP Suite version 6.1.0.29).

## Mechanical property testing

The tensile specimens with gauge dimension of 10 mm (length) × 2.5 mm (width) × 2 mm (thickness) were cut from the as-built blocks[54]. Tensile testing was performed perpendicular to the building direction on an Instron 5584 universal testing machine equipped with an advanced Instron video extensometer at a constant strain rate of 0.001 s$^{-1}$. Four tensile samples were tested for each L-PBF fabricated part with the same addition and heat treatment conditions, and the average value was used as the testing result. After tensile testing, the fracture surface was examined in an SEM. Uniaxial compression tests of the sheet-based gyroid lattices were conducted along the building direction on the same Instron machine. During compression, the samples were centrally located between two pressure plates at a strain rate of 0.001 s$^{-1}$ in accordance with standard ISO13314-2011. The compression process was recorded using a video camera at a rate of 50 frames per second. Frames extracted from these videos were then correlated with features in the associated stress-strain data, providing detailed information regarding the failure modes of both pure Cu and LaB$_6$-doped Cu. All lattices were compressed until densification.

## Electrical conductivity testing

The resistance of pure Cu and 1.0LaB$_6$-Cu was measured on three samples with dimension of 15 mm (length) × 2 mm (width) × 5 mm (height) using a four-point probe method on a ZEM-3 electric resistance measurement system at room temperature. Because the length according to ASTM standard (300 mm) is far beyond the currently available L-PBF machine's capacity to build, the non-standard length was used[56]. The volume resistivity can be calculated as follows:

$$\rho_\nu = (A/l)r \qquad (1)$$

where $\rho_\nu$ is the volume resistivity (Ω mm$^2$ m$^{-1}$), $A$ is the cross-sectional area (mm$^2$), $l$ is the gage length (6 mm), and $r$ is the measured resistance (Ω). The volume resistivity was determined to be 0.019527 Ω mm$^2$ m$^{-1}$ for the L-PBF fabricated pure Cu, and 0.017528 Ω mm$^2$ m$^{-1}$ for 1.0LaB$_6$-Cu. The electrical conductivity $\sigma$ can be expressed as follows:

$$\sigma = \frac{1}{\rho_\nu} = \frac{l}{Ar} \qquad (2)$$

Accordingly, the electrical conductivity was calculated to be $5.121 \times 10^7\,\mathrm{S\,m^{-1}}$ for the pure Cu, and $5.705 \times 10^7\,\mathrm{S\,m^{-1}}$ for $1.0LaB_6$-Cu. In addition, based on ASTM B193, 100% IACS (International Annealed Copper Standard) is defined as the electrical conductivity corresponding to a volume resistivity at 293 K of $0.017241\,\Omega\,\mathrm{mm^2\,m^{-1}}$. Therefore, it is 88.3% IACS for the L-PBF fabricated pure Cu, and 98.4% IACS for $1.0LaB_6$-Cu.

## Thermal conductivity calculation

The thermal conductivity of the L-PBF fabricated pure Cu and $1.0LaB_6$-Cu can be calculated through their electrical conductivity based on the Wiedemann-Franz law[57].

$$\kappa / \sigma = \mathrm{K}L \qquad (3)$$

where $\kappa$ is the thermal conductivity ($\mathrm{W\,m^{-1}\,K^{-1}}$), $\sigma$ is the electrical conductivity ($\mathrm{S\,m^{-1}}$), $K$ is the absolute temperature (K), and $L$ is a Lorenz constant ($\mathrm{W\,\Omega\,K^{-2}}$). For the annealed pure Cu with the electrical conductivity of $5.916 \times 10^7\,\mathrm{S\,m^{-1}}$ and thermal conductivity of $401\,\mathrm{W\,m^{-1}\,K^{-1}}$, Lorenz constant can be calculated to be $2.31 \times 10^{-8}\,\mathrm{W\,\Omega\,K^{-2}}$. Therefore, the thermal conductivity of the L-PBF fabricated pure Cu and $1.0LaB_6$-Cu can be estimated to be $347\,\mathrm{W\,m^{-1}\,K^{-1}}$ and $387\,\mathrm{W\,m^{-1}\,K^{-1}}$, respectively.

## Data availability

The experimental data generated in this study have been deposited in Dryad[58]. Source data are provided with this paper.

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

## Acknowledgements

This work was supported by the Australian Research Council (ARC, grant number: DP210103162). We thank Meng Li of The University of Queensland for undertaking the electrical conductivity test. M.B. acknowledge the funding from Independent Research Fund Denmark, DIGI-3D project (Contract number: 0136–00210B). The authors from Northwestern Polytechnical University thank the State Key Laboratory of Solidification Processing in NWPU for funding support (Grant number: SKLSP202319). The authors from The University of Queensland thank Australian Microscopy & Microanalysis Research Facility at the Centre for Microscopy and Microanalysis (CMM), The University of Queensland for the facilities and technical assistance. The authors are grateful for the scientific and technical support from the Australian Centre for Microscopy and Microanalysis (ACMM) as well as the Microscopy Australia Node at the University of Sydney.

## Author contributions

Y.L., J.Z., C.H. and M.-X.Z. conceived the concept and designed the experiments. M.-X.Z. supervised the project. Y.L. and J.Z. carried out the main experiments. R.N., J.C. and Y.-S.C. conducted the APT characterization. M.B. and J.H.H. contributed idea for the research. Y.Z. conducted data analysis and lattice experiment. Q.T. and J.Z. prepared TEM samples and conducted the TEM characterization. Y.Y., Y.L. and S.L. prepared EBSD samples and carried out the EBSD characterization. Y.L., J.Z., Y.Z., M.L., X.H., M.E., C.H. and M.-X.Z. wrote the manuscript. All authors contributed to the discussion of the data.

## Competing interests

The authors declare no competing interests.
