## [Peer Review File · Nature Communications]

Manufacturing of High Strength and High Conductivity Copper with Laser Powder Bed FusionREVIEWER COMMENTS

Reviewer #1 (Remarks to the Author):

This manuscript provides a study on the effect of lanthanum hexaboride additions on the processability and properties of copper fabricated via laser powder bed fusion. The as-printed mechanical, electrical, and thermal properties of LaB6 doped copper are of some interest. However, I have the following major concerns:

Major Concerns

1. The term '3D Printing' in the title is too broad, please specify the manufacturing technique used in the study.
2. Ln. 38: "remarkable changes in microstructures and properties" What microstructures and properties? Please specify.
3. Ln. 39: "improved L-PBF processability over a much wider processing window" How were processing windows measured in this study?
4. The introduction does a good job at describing the motivation, but I don't understand what the study does or how the experiments were designed. Please include this information in the introduction.
5. Ln. 79-80: Why LaB6 over other potential candidates? Was any comparative analysis done for other particles? What was the methodology for this selection?
6. Section "Mechanical and Conductivity Properties" It is not appropriate to compare highly porous as-printed pure copper and LaB6 doped copper that has higher relative density if the process space has not been fully explored for either of these materials. Anyone can make a highly porous part and compare it to a non-porous part to find improved properties. This is especially true when there is literature that shows significantly lower porosity pure copper parts being fabricated with L-PBF (see ref. below). Their as-printed pure copper showed significantly better properties than those reported in this study (94% IACS, 211 MPa UTS, 43% Elongation):
 - a. Jadhav, S.D., Goossens, L.R., Kinds, Y., Van Hooreweder, B. and Vanmeensel, K., 2021. Laser-based powder bed fusion additive manufacturing of pure copper. *Additive Manufacturing*, 42, p.101990.
7. Ln. 103: "1.0LaB6-Cu can also achieve higher strength." Please quantify and source this statement.
8. Section "Microstructural Characterization" It is critical to first understand porosity and microstructure before mechanical properties are discussed. It is misleading to show large differences in mechanical properties between the doped and non-doped materials without an understanding that one set of materials has severe porosity and the other does not, and that process optimization to reduce porosity has not been done. In fact, I don't think the comparison is appropriate at all. However, I recommend making this the first section.
9. Ln. 152-154: "With an improvement in laser processability, highly dense 1.0LaB6-Cu parts can be fabricated with a much wider L-PBF processing window (Supplementary Fig. 5)." Without an actual exploration of the large multivariate parameter space, I do not see how this claim can be made. Please see the following for an example of a process window comparison between alloy systems for a 3 dimensional subset of the parameter space.
 - a. Seede, R., Ye, J., Whitt, A., Trehern, W., Elwany, A., Arroyave, R. and Karaman, I., 2021. Effect of composition and phase diagram features on printability and microstructure in laser powder bed fusion: Development and comparison of processing maps across alloy systems. *Additive Manufacturing*, 47, p.102258.
10. None of the figures in the manuscript or supplementary figures have quantified scale bars in the images. Please add the numbers to the scale bars so the scale of the figures are clear.
11. Many of the figures have poor resolution. It is difficult to see the text and graphs.
12. Throughout the study it needs to be made clear which "relative density" is being discussed. At which points is the study discussing relative lattice density, and when is it talking about material density/porosity?
13. Ln. 456: "high density cannot be achieved in 0.5LaB6-Cu" How was this verified?
14. Errors in grammar and technical writing in this manuscript make it difficult to read and

understand.

Reviewer #2 (Remarks to the Author):

This manuscript clearly mentions that uniform dispersion of LaB₆ nanoparticles in the copper matrix of L-PBF composite contributes to both high strength and high ductility, as well as high electrical conductivity. On the other hand, the same authors have already clarified the role of LaB₆ nanoparticles as nucleants on microstructures and mechanical properties of L-PBF Al composites in the below published paper.

Q. Ting et al., Demonstrating the roles of solute and nucleant in grain refinement of additively manufactured aluminium alloys, *Additive Manufacturing* 49 (2022) 102516.

Therefore, it is difficult for us to disturb new findings specifically in this manuscript submitted to Nature Communications.

Reviewer #3 (Remarks to the Author):

In today's dynamic technology landscape, the role of 3D printing in producing high-strength, high-conductivity copper is critical. This process combines the remarkable conductivity of copper with the design flexibility of additive manufacturing. Precise microstructure control enables superior mechanical and electrical performance, with implications for the electronics, energy, aerospace and automotive sectors. Lightweight, strong copper components increase efficiency and sustainability, while 3D printing aligns with green manufacturing. This synergy of innovation and material prowess propels us toward an electrifying, sustainable future.

This study successfully employs established analytical techniques, including EBSD, EDS, XRD, FIB-SEM, and APT, to address a relevant research problem. The quality of the data obtained from these commonly used methods attests to the rigor of the analysis performed by the authors. The conclusions drawn from the individual experimental results are coherent and logically sound, although somewhat anticipated. It would be beneficial if these conclusions were synthesized to encapsulate the central message of all the findings. The presentation of a consolidated conclusion, distinct from the detailed results of each experimental phase, could enhance the overall clarity of the paper. The potential significance of the findings is noteworthy. The study effectively fills a current gap in additive copper processing, with significant implications for industrial applications. The methods outlined hold promise for increasing efficiency in the industrial sector and represent a valuable contribution to the field.

In summary, the authors use common analytical tools to address the challenges of additive copper processing. A more concise presentation of the conclusions would enhance the impact of the paper.

The overall structure and graphical presentation need to be substantially revised for better understanding. Due to the broad audience of the journal, some technical details are better presented in the main text rather than in the appendix/supplement to enhance comprehension. Nevertheless, the study is highly significant due to its relevance in addressing an existing gap in the field and its industrial implications. Therefore, this manuscript should undergo a major revision.

General Results: The authors include additional experiments with TiB₂. These experiments are irrelevant to the main results of the paper and it is recommended to delete these passages or move them to the Supplementary. Rather focus on LaB₆.

General to Methods: The methods section should be shortened. For example, information on parameters and machine settings can be abbreviated in tables.

General on figures: The readability of the information in the figures should be checked and improved (font size, assignability of data via color). Displaying the scale bars directly in the figures would be more appropriate.

Page 4 point 2/3: There are 3 theses, of which thesis 2 and 3 are not related to LaB₆. Here the explanation is given for TiB₂. The results or conclusion should emphasize more the hypotheses related

to LaB6.

Page 4, line 81: If particle size is given, it is better to give it in percentiles. For example, d50 or d90.

Page 5, line 96: The reference to Supplementary Note 2 is not conclusive at first glance (use of other wavelengths and not strength. Perhaps this point should be further subdivided).

Page 7, line 150: Since the change in absorbance significantly describes the process and the success of LaB6, a comparison of the absorbance spectrum should be made in the main text. A brief description of why the absorbance is increased and why LaB6 in particular is advantageous here would be good. Is there another way to increase the absorbance?

Page 7, line 159: Were the surfaces at the component boundary also examined? Is there a possibility of agglomeration in this area? Is homogeneity guaranteed even for complex components and very fine structures? Or does this statement only apply to the bulk?

Page 16 Fig. 1: In a) the curves "pure Cu" and "1.0LaB6-Cu-microparticle" are very difficult to distinguish. Please choose another color. In b) the axis labels are too small and the legends (especially the indices) are illegible.

Page 17 Fig. 2: Bringing scale bars into the image and labeling them. The labels (a1,a2,a3 ...) are lost in the false-color image. Please highlight them more clearly. Add a legend for orientation next to the image and enlarge it.

Page 18 Figure 3: Yellow provides insufficient contrast. Please choose another color. The arrows in b) are difficult to see as such. In d), the font size should be adjusted to improve readability.

Page 20, Line 445: Particle size is reported in percentiles.

Page 20 Row 449: Reference 52 is not given.

Supplementary page 4 Figure 1: In e) there is a step in the spectrum at about 840 nm. How can this be explained? It looks very much like a change of sources in the spectrometer and a faulty white balance/calibration of the spectrometer. What part of the spectrum is reliable? Here, the measurement data should be looked at more closely, a correction should be made, or the deviation should be justified.

Supplementary page 13 Figure 10: For b) see previous comment.

Supplementary page 17 Line 190: A typical spot diameter of 200 μ m cannot be given as a general rule. Due to the shorter wavelength, even smaller spots can be achieved than with IR. This is highly dependent on the machine configuration and the process used. For very fine structures, green laser systems are currently being used.

Response to Reviewers' Comments

We are very grateful for the comments from the reviewers and for the Editor providing us with the opportunity to revise our manuscript. We have understood and appreciate the comments of the reviewers and we agree that by addressing their comments, the manuscript has been very substantially improved. We have highlighted all revisions in red in the revised version. In the following sections, the reviewer's comments are in black, our responses to the comments are in blue and red indicates the text added/modified in the manuscript.

Reviewer #1:

This manuscript provides a study on the effect of lanthanum hexaboride additions on the processability and properties of copper fabricated via laser powder bed fusion. The as-printed mechanical, electrical, and thermal properties of LaB₆ doped copper are of some interest.

Response: We are grateful that the reviewer appreciates that the topic is of importance.

1. However, I have the following major concerns:

The term '3D Printing' in the title is too broad, please specify the manufacturing technique used in the study.

Response: We agree with the reviewer. The term "3D printing" has been replaced with "Laser Powder Bed Fusion". The title of the manuscript now reads:

"Manufacturing of High Strength and High Conductivity Copper with Laser Powder Bed Fusion"

2. Ln. 38: "remarkable changes in microstructures and properties" What microstructures and properties? Please specify.

Response: We appreciate that we should have been much more explicit here. This sentence has been re-worded to be more explicit in the revised manuscript, as shown below.

"We show that trace additions of LaB₆ to pure Cu results in an improved L-PBF processability, an enhanced strength, an improved thermal stability, all whilst maintaining a high conductivity"

3. Ln. 39: "improved L-PBF processability over a much wider processing window" How were processing windows measured in this study?

Response: We apologise for this comment. In fact, we really should only say that the L-PBF processability is improved over a wider range of 'laser powers', with the addition of LaB₆ as shown in Supplementary Fig. 3. We did not modify the scanning speed, hatch spacing, layer thickness, or other parameters of L-PBF, so it is not appropriate to assert a much wider 'processing window'. We appreciate the reviewer for pointing this out. We have revised the relevant statement on Page 5, as shown below.

"...highly dense 1.0LaB₆-Cu parts can be fabricated effectively over a broader laser power range, spanning 325 W to 400 W"

4. The introduction does a good job at describing the motivation, but I don't understand what the study does or how the experiments were designed. Please include this information in the introduction.

Response: We appreciate the reviewer for this comment that we have not articulated well the main innovation of our approach. To do a better job of this, we have revised sections on Page 3 to better clarify the objectives of the study and to provide a better understanding of the new innovation in our design, as shown below:

“To address the challenge posed by the limited laser processability of pure Cu and the trade-off between strength and conductivity or ductility, here we present a new design strategy for L-PBF of pure Cu. We select a particular additive to mix with pure Cu powder based on this new design strategy. The key feature of our approach is that we select additives, of which the constituent elements have negligible solubility in solid Cu (and therefore have negligible detrimental effects on the conductivity), but will dissolve in the melt pools during laser melting (therefore not too high melting point) and subsequently re-precipitate with a very fine dispersion during solidification (and hence provide excellent strengthening). This is a new approach and criteria to selecting additions to help L-PBF of pure Cu and as we show in this contribution, it works extremely well and allows us to access a new combination of printability, strength, conductivity and thermal stability.”

5. Ln. 79-80: Why LaB₆ over other potential candidates? Was any comparative analysis done for other particles? What was the methodology for this selection?

Response: The key factors driving the selection of LaB₆ are the negligible solubility of La and B in the solid Cu, which is critical in maintaining the high conductivity of the Cu, and the comparatively low melting point of LaB₆ (2,210°C), compared to other carbides, nitrides, and borides that might be considered for this purpose (as detailed in Supplementary Table 3). It is critical that the LaB₆ does dissolve in the melt pool, otherwise it is not possible to get it to re-precipitate during solidification with the fine nanoscale distribution that is necessary to obtain the strong strengthening increment. If the LaB₆ did not dissolve, then the LaB₆ particles observed in the consolidated material would be large and the strengthening increment minimal. LaB₆, like other carbides, nitrides, and borides improve the laser absorptivity, but it also has a low wettability angle with the Cu which helps the uniform dispersion. The LaB₆ addition, is really a quite unique and special one for the case of pure Cu, as we see from the combinations of properties exhibited, but the design approach can be applied to other alloys, where other additions will likely be more suited.

We did conduct a parallel experiment involving the L-PBF fabrication of 1.0TiB₂-Cu. TiB₂ has a significantly higher melting point than LaB₆, and Ti also exhibits solubility in Cu. The resulting L-PBF material is well consolidated but lower conductivity (because of Ti in solid solution) and lower strength because the TiB₂ did not fully dissolve and re-precipitate so the TiB₂ particles are similar size to the original additions. This TiB₂ example proves our hypothesis and the value of our new design approach in selecting additions – which is the key innovation in this work.

To try and make this clearer, we have revised the Supplementary Note 1, specifically focusing on the selection of the LaB₆ addition, as shown below.

Supplementary Note 1 – Selection of additive.

Within the category of commonly used ceramics, encompassing oxides, carbides, nitrides, and borides (Supplementary Table 3), LaB₆ stands out due to its notably lower melting point of 2,210°C. LaB₆ has previously found application as a grain refiner in additive manufacturing for aluminium alloys⁷, yet its utilization in Cu remains largely unexplored. La and B exhibit negligible solid solubility in Cu, which means any solute trapping in the solid can be minimised and detrimental effects on the conductivity can be minimised. The low melting point (compared to other possible additions) improves the potential for complete dissolution of the particles in the melt pools, followed by subsequent re-precipitation during solidification (due to the negligible solid solubility of La and B). This is the key new design idea (dissolution of the LaB₆ and complete re-precipitation) that drives this work. LaB₆ also possesses a relatively low wetting angle of 71° when contacting with liquid Cu, affirming that it fulfils the proposed selection criteria.

Furthermore, we have added the parallel experiment on TiB₂ additions in the supplementary information as Supplementary Note 7, as shown below.

“Supplementary Note 7 – Parallel experiment of the L-PBF fabricated 1.0TiB₂-Cu.

To further validate our design strategy, we carried out a parallel experiment using TiB₂ nanoparticles with the same particle size and addition level (Methods). Compared to LaB₆, TiB₂-doped Cu powder feedstock also has a higher laser absorptivity (Supplementary Fig. 13a, b) and hence the as-fabricated 1.0TiB₂-Cu part exhibits high density without any lack-of-fusion defects (Supplementary Fig. 13c). However, the SEM examination demonstrates that, unlike the LaB₆, agglomeration of TiB₂ nanoparticles took place along the grain boundaries (Supplementary Fig. 13d). This is due to the high wetting angle between the TiB₂ and Cu melt (Supplementary Table 3). Furthermore, the relatively high melting point of TiB₂ led to incomplete melting of the particles. The solid solubility of Ti in Cu inhibits the re-precipitation of TiB₂ from the melt and reduce the conductivity. Therefore, the 1.0TiB₂-Cu only achieves an ultimate tensile strength of 252 ± 4 MPa (Supplementary Fig. 13e) and an electrical conductivity of 91.2% IACS, which are much lower than the 1.0LaB₆-Cu part. This confirms that all the selection criteria of additives are essential to produce high performance Cu parts. Given that only the commonly used ceramics were considered in this work, it is expected that other intermetallic compounds and/or other ceramics can be sought to meet the selection criteria for a range of different metals.”

6. Section “Mechanical and Conductivity Properties” It is not appropriate to compare highly porous as-printed pure copper and LaB₆ doped copper that has higher relative density if the process space has not been fully explored for either of these materials. Anyone can make a highly porous part and compare it to a non-porous part to find improved properties. This is especially true when there is literature that shows significantly lower porosity pure copper parts being fabricated with L-PBF (see ref. below). Their as-printed pure copper showed significantly better properties than those reported in this study (94% IACS, 211 MPa UTS, 43% Elongation):

a. Jadhav, S.D., Goossens, L.R., Kinds, Y., Van Hooreweder, B. and Vanmeensel, K., 2021. Laser-based powder bed fusion additive manufacturing of pure copper. Additive Manufacturing, 42, p.101990.

Response: We acknowledge the reference provided by the reviewer, which demonstrates the possibility of 3D printing highly dense pure copper using an infrared laser at 500 W power. The reference has been cited in the revised manuscript. We agree with the reviewer that it is

not proper to compare porous materials with highly dense materials. Hence, we now compare the strength and conductivity of copper produced in our work with a number of fully dense copper alloys in published work on pages 8 and 9. We noticed that although the high-density pure copper produced with high power may have high conductivity, the yield strength remains low at 122 MPa. We did briefly mention the properties of additively manufactured pure copper with high density in the main text and supplementary information. Highly dense pure copper components can be produced using smaller laser wavelengths, such as green laser, in comparison to infrared laser. As shown in Supplementary Fig. 11, the strength of such parts ranges from 127 to 180 MPa, which is slightly higher than that achieved using high-power technology in Jadhav's work. However, it is still much lower than our work, which is 347 MPa.

We have added several sentences in the main text on Page 8, so that this key point is made clearer, as shown below.

“The low strength and conductivity of the pure Cu are due to the presence of the high density of pores.”

“While it is possible to create highly dense pure Cu parts through AM using green laser, the strength of these parts remains significantly lower than that achieved in our study^{15,16}. The significant strength enhancement is mainly attributed to dispersion strengthening from the LaB₆ nanoparticles”

Additionally, we have restructured the manuscript, as suggested by the reviewer in Comment 8. We've moved the "Microstructural Characterization" section to the beginning of the results section. Subsequently, the "Mechanical and Conductivity Properties" section now follows the microstructural analysis, aligning with the reviewer's recommendation.

7. Ln. 103: “1.0LaB₆-Cu can also achieve higher strength.” Please quantify and source this statement.

Response: As documented in Reference 1, GRCop-42 and GRCop-84 alloys produced via L-PBF displayed a yield strength ranging from 172 to 208 MPa¹. In contrast, the 1.0LaB₆-Cu alloy exhibits a significantly higher yield strength of 347 MPa.

To address this comment, one sentence has been revised on Page 9, as shown below.

“...the 1.0LaB₆-Cu also exhibits higher strength compared to GRCop-42 and GRCop-84 alloys, with the latter alloys demonstrating a yield strength within the range of 172 to 208 MPa⁴⁰”

References for Response to comment 7:

[1] Gradl, P. R. *et al.* in *AIAA Propulsion and Energy 2019 Forum AIAA Propulsion and Energy Forum* (American Institute of Aeronautics and Astronautics, 2019).

8. Section “Microstructural Characterization” It is critical to first understand porosity and microstructure before mechanical properties are discussed. It is misleading to show large differences in mechanical properties between the doped and non-doped materials without an understanding that one set of materials has severe porosity and the other does not, and that process optimization to reduce porosity has not been done. In fact, I don't think the comparison is appropriate at all. However, I recommend making this the first section.

Response: We appreciate and agree with this suggestion and have restructured the manuscript. The "Microstructural Characterization" section has been moved to the beginning of the results section, and the "Mechanical and Conductivity Properties" section now follows.

9. Ln. 152-154: "With an improvement in laser processability, highly dense 1.0LaB₆-Cu parts can be fabricated with a much wider L-PBF processing window (Supplementary Fig. 5)." Without an actual exploration of the large multivariate parameter space, I do not see how this claim can be made. Please see the following for an example of a process window comparison between alloy systems for a 3 dimensional subset of the parameter space.

a. Seede, R., Ye, J., Whitt, A., Trehern, W., Elwany, A., Arroyave, R. and Karaman, I., 2021. Effect of composition and phase diagram features on printability and microstructure in laser powder bed fusion: Development and comparison of processing maps across alloy systems. Additive Manufacturing, 47, p.102258.

Response: We agree with the reviewer. Our comment on a "much wider processing window" is not justified since we only examined variations in laser power. We have made changes to the text as outlined in our response to Comment 3.

10. None of the figures in the manuscript or supplementary figures have quantified scale bars in the images. Please add the numbers to the scale bars so the scale of the figures are clear.

Response: We apologise for this scale bar issue. Numeric scale bars have been incorporated into all the figures, ensuring the clarity of the figure scales.

11. Many of the figures have poor resolution. It is difficult to see the text and graphs.

Response: We are grateful to the reviewer for bringing the image quality issue to our attention. The font size of text and the figures have been enlarged to enhance readability. The uploaded figures are of high resolution and we apologise for any downsizing that has occurred in the processing of the manuscript.

12. Throughout the study it needs to be made clear which "relative density" is being discussed. At which points is the study discussing relative lattice density, and when is it talking about material density/porosity?

Response: We now use term "relative lattice density" to refer to the lattice density, and "part density" has been employed to describe the density of the L-PBF fabricated solid part.

13. Ln. 456: "high density cannot be achieved in 0.5LaB₆-Cu" How was this verified?

Response: A part density of 97.94% was achieved in the micro-computed tomography (micro-CT) analysis of 0.5LaB₆-Cu, as illustrated in Supplementary Fig. 2c. This result has also been confirmed through Archimedes' method.

14. Errors in grammar and technical writing in this manuscript make it difficult to read and understand.

Response: We are very sorry for the readability. The manuscript has undergone an additional round of editing by our native English-speaking co-authors. We dearly hope the readability is now improved.

Reviewer #2:

This manuscript clearly mentions that uniform dispersion of LaB₆ nanoparticles in the copper matrix of L-PBF composite contributes to both high strength and high ductility, as well as high electrical conductivity. On the other hand, the same authors have already clarified the role of LaB₆ nanoparticles as nucleants on microstructures and mechanical properties of L-PBF Al composites in the below published paper.

Q. Ting et al., Demonstrating the roles of solute and nucleant in grain refinement of additively manufactured aluminium alloys, Additive Manufacturing 49 (2022) 102516.

Therefore, it is difficult for us to disturb new findings specifically in this manuscript submitted to Nature Communications.

Response: Thank you for this comment. We see that our new design idea that is the core innovation of the present paper, and how it differs from previous work, has not been made clear enough. This is related to our response to Comment 4 of Reviewer #1.

The first and most important point is that we are not using LaB₆ as a grain refining nucleant in Cu. It can be used for this purpose for low melting point aluminium but this is not the key mechanism in Cu, nor the basis for its choice. If grain refinement was all that was important, we could just use TiB₂ (as discussed in our response to Comments 5 from Reviewer #1) and whilst this may allow dense Cu to be prepared by L-PBF, the strength and conductivity are both poor (these results are shown in the Supplementary Note 7).

The key feature of our new approach is that we select additives that have negligible solubility in solid Cu (and hence have negligible detrimental effects on the conductivity), but which will dissolve in the melt pool during laser melting (therefore not too high melting point) but subsequently re-precipitate with a very fine dispersion during solidification (and hence provide excellent strengthening). This is a new approach and criteria to selecting additions to help L-PBF of pure Cu and as we show in this contribution, it works extremely well and allows us to access a new combination of printability, strength, conductivity and thermal stability.

The key factors driving the selection of LaB₆ are the negligible solubility of La and B in the solid Cu, which is critical in maintaining the high conductivity of the Cu, and the comparatively low melting point of LaB₆ (2,210°C), compared to other carbides, nitrides, and borides that might be considered for this purpose (as detailed in Supplementary Table 3). It is critical that the LaB₆ does dissolve in the melt pool, otherwise it is not possible to get it to re-precipitate during solidification with the fine nanoscale distribution that is necessary to obtain the strong strengthening increment. If the LaB₆ did not dissolve, then the LaB₆ particles observed in the consolidated material would be large and the strengthening increment minimal (as we see with TiB₂ additions). LaB₆, like other carbides, nitrides, and borides improve the laser absorptivity, but it also has a low wettability angle with the Cu which helps the printing. The LaB₆ addition, is really a quite unique and special one for the case of pure Cu, as we see from the properties exhibited, but the design approach can be applied to other alloys, where other additions will likely be more suited.

The red text in our response to Comment 4 of Reviewer #1 has been added to the manuscript text to more clearly articulate this fundamental difference in the design strategy used in this work, and the rationale for choosing LaB₆, compared to previous works simply using LaB₆ as a grain refining agent.

Reviewer #3:

In today's dynamic technology landscape, the role of 3D printing in producing high-strength, high-conductivity copper is critical. This process combines the remarkable conductivity of copper with the design flexibility of additive manufacturing. Precise microstructure control enables superior mechanical and electrical performance, with implications for the electronics, energy, aerospace and automotive sectors. Lightweight, strong copper components increase efficiency and sustainability, while 3D printing aligns with green manufacturing. This synergy of innovation and material prowess propels us toward an electrifying, sustainable future. This study successfully employs established analytical techniques, including EBSD, EDS, XRD, FIB-SEM, and APT, to address a relevant research problem. The quality of the data obtained from these commonly used methods attests to the rigor of the analysis performed by the authors. The conclusions drawn from the individual experimental results are coherent and logically sound, although somewhat anticipated. It would be beneficial if these conclusions were synthesized to encapsulate the central message of all the findings. The presentation of a consolidated conclusion, distinct from the detailed results of each experimental phase, could enhance the overall clarity of the paper. The potential significance of the findings is noteworthy. The study effectively fills a current gap in additive copper processing, with significant implications for industrial applications. The methods outlined hold promise for increasing efficiency in the industrial sector and represent a valuable contribution to the field. In summary, the authors use common analytical tools to address the challenges of additive copper processing. A more concise presentation of the conclusions would enhance the impact of the paper. The overall structure and graphical presentation need to be substantially revised for better understanding. Due to the broad audience of the journal, some technical details are better presented in the main text rather than in the appendix/supplement to enhance comprehension. Nevertheless, the study is highly significant due to its relevance in addressing an existing gap in the field and its industrial implications. Therefore, this manuscript should undergo a major revision.

Response: We are grateful the reviewer appreciates the importance of the work and the topic.

As we outline in our responses to Reviewer #1 Comment 4, and to Reviewer #2, our design approach which involves selecting additions that dissolve and then re-precipitate is quite different to any other strategies currently used in for similar additions in L-PBF (which are basically designed for grain nucleants), but we also appreciate we have not made this key innovation clear enough in the paper.

By adding the **red text** included in our response to Reviewer #1 Comment 4, we hope that we capture the key innovation and conclusions more concisely.

1. General Results: The authors include additional experiments with TiB₂. These experiments are irrelevant to the main results of the paper and it is recommended to delete these passages or move them to the Supplementary. Rather focus on LaB₆.

Response: We have relocated the parallel experiment of the L-PBF fabricated 1.0TiB₂-Cu and the related discussion to Supplementary Note 7 in the supplementary information.

2. General to Methods: The methods section should be shortened. For example, information on parameters and machine settings can be abbreviated in tables.

Response: We agree and appreciate this suggestion. We have shortened the method section, by incorporating parameters and machine settings for L-PBF into tables, as demonstrated below.

Supplementary Table 1 | L-PBF processing parameters.

L-PBF processing parameters	Value	Unit
Laser power	375	W
Scanning speed	400	mm s ⁻¹
Layer thickness	30	μm
Hatch spacing	120	μm
Spot size	80	μm
Preheating temperature	200	°C
Oxygen concentration	<0.05	vol%

3. General on figures: The readability of the information in the figures should be checked and improved (font size, assignability of data via color). Displaying the scale bars directly in the figures would be more appropriate.

Response: We have enhanced the readability of the figures by increasing the font size and improving the colour contrast. Numeric scale bars have been incorporated into all the figures, ensuring the clarity of the figure scales.

4. Page 4 point 2/3: There are 3 theses, of which thesis 2 and 3 are not related to LaB₆. Here the explanation is given for TiB₂. The results or conclusion should emphasize more the hypotheses related to LaB₆.

Response: Among commonly used ceramics such as oxides, carbides, nitrides and borides (Supplementary Table 3), LaB₆ exhibits a much lower melting point of 2,210 °C. Specifically, its constituent elements, namely La and B, show extremely low solid solubility in Cu. This makes possible to fully dissolve the particles into the melt pools followed by re-precipitation during solidification without significant solute trapping in the solid, and therefore minimize the conductivity reduction caused by solid solution effect. In addition, the wetting angle of molten Cu on LaB₆ is 71°, which is relatively low to meet the proposed selection criteria.

To address this comment, the selection of additive has been presented in Supplementary Note 1 and a couple of sentences have been added in the main text on Page 4, as shown below.

“Following this design strategy, we identified LaB₆ as a suitable additive candidate. LaB₆ met the criteria due to its lower melting point, minimal solid solubility of constituent elements in Cu, and low wetting angle with molten Cu (see Supplementary Note 1)”

5. Page 4, line 81: If particle size is given, it is better to give it in percentiles. For example, d50 or d90.

Response: The percentiles of LaB₆ powder are now given on Page 22 in the revised manuscript, as shown below.

“The LaB₆ powder demonstrates D10, D50, and D90 measurements of 54 nm, 85 nm, and 145 nm, respectively”

6. Page 5, line 96: The reference to Supplementary Note 2 is not conclusive at first glance (use of other wavelengths and not strength. Perhaps this point should be further subdivided.

Response: The figure in Supplementary Note 5 of the revised supplementary information has been updated to use yield strength in the legend instead of wavelengths, as shown below.

Supplementary Fig. 12 | Comparison of tensile properties of AM fabricated pure Cu using green laser and electron beam with that of 1.0LaB₆-Cu.

7. Page 7, line 150: Since the change in absorbance significantly describes the process and the success of LaB₆, a comparison of the absorbance spectrum should be made in the main text. A brief description of why the absorbance is increased and why LaB₆ in particular is advantageous here would be good. Is there another way to increase the absorbance?

Response: Certainly, the absorbance is important in the consolidation during printing, but there are many different additions that could do this. Absorbance is not the only key role of the LaB₆ in this study. It is a necessary, but not sufficient criteria. The LaB₆ is also unique in that the La and B have negligible solid solubility, and the melting point of LaB₆ is sufficiently low that it can dissolve in the melt pool and then re-precipitate as a fine dispersion (which is key to the high strength and good conductivity). It is this combination of reasons, not only the absorbance, that determines the choice of LaB₆.

Regarding the absorbance, following the mechanical mixing of copper powder with 1.0 wt% LaB₆ nanoparticles, LaB₆ nanoparticles can be uniformly coated on the surface of the copper particles (Supplementary Fig. 1). This homogenous distribution is advantageous because LaB₆ has significantly lower laser reflectivity than pure copper, as demonstrated in Fig. 1b. Consequently, the Cu powder with mixing of LaB₆ nanoparticles possesses improved laser absorptivity. Moreover, the presence of LaB₆ nanoparticles increases the surface roughness of copper powder, aiding laser absorption by promoting multiple reflections within the powder bed^{2,3}.

To address this comment, the laser reflectivity test results have been moved to Fig. 1 in the main text, as shown below.

Fig. 1 | Laser AM of Cu and LaB₆-doped Cu.

Additionally, we have added the relevant discussion on Page 4 of the revised manuscript, as shown below.

“The laser reflectivity test clearly demonstrates that pure Cu powder exhibits exceptionally high laser reflectivity within the infrared laser range (Fig. 1b), specifically at a wavelength of 1,060 nm, as used by the laser powder bed fusion (L-PBF) system in our study. The reflectivity was noticeably decreased upon the incorporation of 1 weight per cent (wt%) LaB₆ nanoparticles. This reduction can be attributed to two factors: the inherently low laser reflectivity of LaB₆ and the introduction of LaB₆ nanoparticles, which enhance the surface roughness of pure Cu particles, facilitating multiple reflections within the powder bed.”

In addition to the method proposed in this study, green laser beams and electron beams are also utilized for 3D printing of high-density pure copper parts. To emphasize the novelty of our current work, we have provided a comparison of our method with others, which is presented in Supplementary Note 5.

References for Response to comment 7:

- [2] Li, X. P. *et al.* Selective laser melting of nano-TiB₂ decorated AlSi10Mg alloy with high fracture strength and ductility. *Acta Mater.* **129**, 183-193 (2017).
- [3] Ang, L. K., Lau, Y. Y., Gilgenbach, R. M., & Spindler, H. L. Analysis of laser absorption on a rough metal surface. *Appl. Phys. Lett.* **70**, 696-698 (1997).

8. Page 7, line 159: Were the surfaces at the component boundary also examined? Is there a possibility of agglomeration in this area? Is homogeneity guaranteed even for complex components and very fine structures? Or does this statement only apply to the bulk?

Response: We really appreciate this question. We examined the top surface of the as L-PBF fabricated sample, and observed uniformly distributed nanoparticles. The results from the single-track experiment also confirmed the uniform distribution of these nanoparticles. It demonstrates that the proposed design strategy is well-suited for geometrically complicated components with very fine structures.

To address this comment, we have incorporated the SEM image depicting nanoparticles on the top surface of the L-PBF fabricated 1.0LaB₆-Cu into the Supplementary Fig. 8 of the revised supplementary information on Page 11. Accordingly, we have also provided a short discussion in the first paragraph of the main text on Page 7.

Supplementary Fig. 8 | Nanoparticles in the top surface of the L-PBF fabricated 1.0LaB₆-Cu.

“In addition, we also examined the top surface of the L-PBF fabricated 1.0LaB₆-Cu sample, where there is no extensive thermal cycling, and as we observed in other areas, we also found a large number of uniformly distributed LaB₆ nanoparticles (Supplementary Fig. 8). This observation, coupled with the results from the single-track experiment, suggests that the proposed design strategy is suitable for the geometrically complex components with fine structures due to the formation of uniformly dispersed nanoparticles during solidification.”

9. Page 16 Fig. 1: In a) the curves "pure Cu" and "1.0LaB₆-Cu-microparticle" are very difficult to distinguish. Please choose another color. In b) the axis labels are too small and the legends (especially the indices) are illegible.

Response: We have changed the colour to increase the contrast, and we've increased the font size in Fig. 4 of the revised manuscript to enhance readability, as shown below.

Fig. 4 | Tensile mechanical property and electrical conductivity.

10. Page 17 Fig. 2: Bringing scale bars into the image and labeling them. The labels (a1,a2,a3 ...) are lost in the false-color image. Please highlight them more clearly. Add a legend for orientation next to the image and enlarge it.

Response: We have now added numeric scale bars to all the figures to make the scale clear. Additionally, we've adjusted the labels in the figures to enhance readability, as shown in the Response to Comment 7.

11. Page 18 Figure 3: Yellow provides insufficient contrast. Please choose another color. The arrows in b) are difficult to see as such. In d), the font size should be adjusted to improve readability.

Response: We apologise for these poor colour choices. We have replaced yellow with high-contrast white, and increased the font size in Fig. 2 of the revised manuscript to enhance readability, as shown below.

Fig. 2 | Characterization of LaB₆ nanoparticles.

12. Page 20, Line 445: Particle size is reported in percentiles.

Response: The percentiles of powders are now given on Page 22 in the revised manuscript, as shown below.

“The Cu powder exhibits D10, D50, and D90 values of 1 μm, 10 μm, and 64 μm, respectively”

“The LaB₆ powder demonstrates D10, D50, and D90 measurements of 54 nm, 85 nm, and 145 nm, respectively”

13. Page 20 Row 449: Reference 52 is not given.

Response: We apologise for this important omission. In the revised manuscript, we have included Reference 52, as shown below.

“52. Karabulut, Y. & Ünal, R. Additive manufacturing of ceramic particle-reinforced aluminum-based metal matrix composites: a review. *Journal of Materials Science* **57**, 19212-19242, (2022).”

14. Supplementary page 4 Figure 1: In e) there is a step in the spectrum at about 840 nm. How can this be explained? It looks very much like a change of sources in the spectrometer and a

faulty white balance/calibration of the spectrometer. What part of the spectrum is reliable? Here, the measurement data should be looked at more closely, a correction should be made, or the deviation should be justified.

Response: We have conducted multiple tests on the laser reflectivity of both pure copper and powder mixtures. Consistently, our results displayed a prominent step around the 840 nm wavelength, which has been previously documented⁴. To be forthright, we do not currently have an explanation for this phenomenon. However, it is worth noting that our primary focus lies in the laser reflectivity at a wavelength of 1,060 nm, which falls within the infrared laser range. It has been reported that the reflectivity of copper powder for infrared laser fluctuates between 74% to 79%, depending on the particle size⁵⁻⁷. In our work, we measured the laser reflectivity of pure copper powder and found it to be 78% at a wavelength of 1,060 nm. This result closely aligns with the reported values, reinforcing the reliability of our test outcomes.

References for Response to comment 14:

- [4] <https://www.mrj-lasermark.com/info/difficulties-in-3d-printing-of-copper-alloys-a-72961830.html>
- [5] Jadhav, S. D. *et al.* Surface modified copper alloy powder for reliable laser-based additive manufacturing. *Addit. Manuf.* **35**, 101418 (2020).
- [6] Silbernagel, C. *et al.* Electrical resistivity of pure copper processed by medium-powered laser powder bed fusion additive manufacturing for use in electromagnetic applications. *Addit. Manuf.* **29**, 100831, (2019).
- [7] Jadhav, S. D., Goossens, L., Kinds, Y., Hooreweder, B. V. & Vanmeensel, K. Laser-based powder bed fusion additive manufacturing of pure copper. *Addit. Manuf.*, 101990, (2021).

15. Supplementary page 13 Figure 10: For b) see previous comment.

Response: Please refer to the Response to the Comment 14.

16. Supplementary page 17 Line 190: A typical spot diameter of 200µm cannot be given as a general rule. Due to the shorter wavelength, even smaller spots can be achieved than with IR. This is highly dependent on the machine configuration and the process used. For very fine structures, green laser systems are currently being used.

Response: We agree with the reviewer. While the present reports mentioned a green laser beam spot diameter larger than 200 µm, it is indeed possible to achieve smaller spot sizes. As a result, the discussion regarding spot diameter has been omitted from the revised supplementary information.

REVIEWERS' COMMENTS

Reviewer #1 (Remarks to the Author):

The revisions to the manuscript are thorough and acceptable to me. Publication of this manuscript will be of interest to both industry and academia. I have only one more concern listed below:

1. It would be useful to add the mechanical and electrical properties observed in high relative density LPBF fabricated pure copper (94% IACS, 211 MPa UTS, 43% Elongation) to Fig. 4 from the following reference:

a) Jadhav, S.D., Goossens, L.R., Kinds, Y., Van Hooreweder, B. and Vanmeensel, K., 2021. Laser-based powder bed fusion additive manufacturing of pure copper. *Additive Manufacturing*, 42, p.101990.

Response to Reviewers' Comments

We are very grateful for the comments from the editor and reviewers. In the following sections, the editor and reviewer's comments are in black, our responses to the comments are in blue and red indicates the text added/modified in the manuscript.

Reviewer #1:

The revisions to the manuscript are thorough and acceptable to me. Publication of this manuscript will be of interest to both industry and academia. I have only one more concern listed below:

1. It would be useful to add the mechanical and electrical properties observed in high relative density LPBF fabricated pure copper (94% IACS, 211 MPa UTS, 43% Elongation) to Fig. 4 from the following reference:

a) Jadhav, S.D., Goossens, L.R., Kinds, Y., Van Hooreweder, B. and Vanmeensel, K., 2021. Laser-based powder bed fusion additive manufacturing of pure copper. Additive Manufacturing, 42, p.101990.

Response: We thank the reviewer for the comment. The mechanical and electrical properties sourced from the recommended reference (Ref. 13 in the revised manuscript) have been incorporated into Fig. 4, as shown below.

Fig. 4 | Tensile mechanical property and electrical conductivity.

Reviewer #3:

1. Why was Supplementary Fig. 1 panel e removed? Please re-introduce this in the final version for transparency and explain the laser reflectivity in the discussion.

Response: In reply to this reviewer's original Comment 7 "Since the change in absorbance significantly describes the process and the success of LaB₆, a comparison of the absorbance spectrum should be made in the main text", the original Supplementary Fig. 1e has been moved to Fig. 1b within the main text as shown below.

Fig. 1 | Laser AM of Cu and LaB₆-doped Cu.

Additionally, we have added the relevant discussions on Page 4 of the 1st revised version of the manuscript, as shown below.

“The laser reflectivity test clearly demonstrates that pure Cu powder exhibits exceptionally high laser reflectivity within the infrared laser range (Fig. 1b), specifically at a wavelength of 1,060 nm, as used by the laser powder bed fusion (L-PBF) system in our study. The reflectivity was noticeably decreased upon the incorporation of 1 weight per cent (wt%) LaB₆ nanoparticles. This reduction can be attributed to two factors: the inherently low laser reflectivity of LaB₆ and the introduction of LaB₆ nanoparticles, which enhance the surface roughness of pure Cu particles, facilitating multiple reflections within the powder bed.”

2. Please explain the small jump in reflectivity associated with Supplementary Fig. 13 in the discussion.

Response: In appreciation of this comment, the small jump in reflectivity associated with Supplementary Fig. 13 is discussed on Page 23 of the currently revised supplementary information, as shown below.

“Our laser reflectivity test exhibits a prominent step around the 840 nm wavelength (Supplementary Fig. 13b). Although this has been previously reported and documented³³, the actual causes are unclear. Because our current focus lies in the laser reflectivity at a wavelength of 1,060 nm, as used by the L-PBF system in our study, we may have to leave this phenomenon for future study. The measured laser reflectivity of pure Cu at wavelength of 1,060 nm is 78%, which closely aligns with the reported values in the range of 74%-79%^{27,34,35}.”